# The Role of Contextual Variables and Structural Diversity on College Students’ Engineering Self-Efficacy

**DOI:** 10.3390/bs14070564

**Published:** 2024-07-04

**Authors:** Bo Hyun Lee, Xiaotian Hu, Lisa Y. Flores, Rachel L. Navarro

**Affiliations:** 1Educational Studies, College of Education and Human Ecology, The Ohio State University, Columbus, OH 43210, USA; 2Department of Educational, School & Counseling Psychology, College of Education & Human Development, University of Missouri, Columbia, MO 65201, USA; xhrgd@mail.missouri.edu; 3Psychological Science, College of Arts and Science, University of Missouri, Columbia, MO 65201, USA; floresly@missouri.edu; 4Education, Health & Behavior Studies, College of Education & Human Development, University of North Dakota, Grand Forks, ND 58202, USA; rachel.navarro@und.edu

**Keywords:** underrepresented racial minority (URM) faculty, structural equity, engineering, underrepresented racial/ethnic minority, perceived barriers and supports

## Abstract

Structural diversity is defined as the numerical representation of diverse racial/ethnic student groups on campus as one way of exposing students to diversity in higher education. The current study implemented the concept of structural diversity on faculty in higher education, given the significant and unique roles in STEM education. We integrated the proportion of URM faculty within the College of Engineering as a moderating variable in the social cognitive career theory (SCCT) model. With a sample of 254 diverse engineering students from six universities, the results indicated that both perceived engineering barriers and perceived engineering supports significantly related to perceived self-efficacy even after controlling for the effects of the other. Perceived engineering supports mediated the effects of engineering barriers on self-efficacy. Moreover, a moderated mediation effect by the proportion of URM faculty was observed, showing that when the proportion of URM faculty reached a certain level, high levels of perceived engineering barriers had no effect on increasing perceived engineering supports. Implications for fostering career development in engineering with a systematic-tailored approach are discussed.

## 1. Introduction

Diversity, equity, and inclusion (DEI) in higher education are essential to achieving fairness in society. One of the goals for many institutions of higher education in the United States (U.S.) has been diversifying the undergraduate student body and retaining a diverse cohort to graduation. However, this has been an elusive goal, particularly in STEM (science, technology, engineering, and mathematics) fields. Although the U.S. is projected to continue becoming a more racially and ethnically pluralistic society [1], the share of STEM degrees awarded to individuals from underrepresented minority groups (URM; i.e., Hispanics or Latinx, Blacks or African Americans, American Indians or Alaska Natives, Native Hawaiian or Other Pacific Islander, and individuals from more than one race) has remained steady over the past two decades [2]. For example, only 26% of STEM degree holders and 24% of STEM workers were identified as URM in 2020, compared with 20% and 18% in 2011, respectively. Considering that URMs make up 46% of the U.S. population aged between 18 and 34 years old, this is a significantly low proportion of representation in STEM. Among the various STEM fields, engineering is one of the least integrated professions. Historically and currently, engineering is dominated by White, middle- and upper-class men. Specifically, Latinx American, African American, Native American, Native Hawaiian, and biracial/multiracial individuals earned 12%, 4%, 0.3%, 0.1%, and 4% of undergraduate engineering degrees in 2022, whereas 49% of degrees were earned by their White counterparts [3].

### 1.1. Benefits of Diverse Environments in Engineering

Currently, several states have introduced legislation to restrict or eliminate DEI funding, practices, and programs that impact public higher education systems. According to the Chronicle of Higher Education [4], a total of 85 bills have been introduced in 28 states across the U.S. and in Congress that would eliminate DEI offices, mandatory DEI training, diversity statements, and identity-based hiring and admissions practices; 12 states have signed anti-DEI legislation into law. Efforts to restrict DEI practices persist in spite of the numerous benefits to students of exposure to diversity and racially diverse environments that have been documented. According to Wong [5], racially diverse social connections can provide valuable social capital given that “ties to dissimilar others provide access to non-redundant information, resources, and opportunities” (p. 1). Scholars have documented positive academic, intellectual, and social development during college that was facilitated by cross-racial interactions [6,7,8], including improved retention and persistence rates [9], stronger interest in intellectual engagement [10], increased comfort with people of different backgrounds [11], improved overall satisfaction with college [6,7], and expanded cultural awareness, cross-cultural empathy, and social perspective taking [12,13,14]. Workplace productivity and innovation also support the positive outcomes of diversity. For example, as a way of unlocking innovation and driving market growth, Hewlett et al. [15] suggested diversity within the workplace. According to their report, there is a 152% likelier capacity to understand end users by the entire team when at least one member of a team has a shared ethnicity with clients than the counterpart (i.e., a team who does not share the end users’ ethnicity). Given that all these outcomes are significant in both STEM education and the workforce [16], it is critical to facilitate conditions where engineering students are immersed in a diverse environment.

Among multiple factors that have been identified as indicators to influence students’ perceptions of a positive campus climate and respect for diversity, faculty diversity has gained national attention [17,18]. In addition to a wide array of benefits from the high quality of student–faculty interactions, scholars have argued that increased faculty diversity would mark up advantages for underrepresented students, including narrowing down the academic achievement gap between white and underrepresented students [19] and a better higher percentage of recruitment and retention of underrepresented students [8]. In STEM, Price [20] documented a positive correlation by Black identified students between the persistence and the number of STEM courses taught by a Black instructor, suggesting an integral role of underrepresented racial minority (URM) faculty in helping to shape underrepresented students’ sense of belonging through their interactions [21]. As such, Syed et al. [22] named faculty as an identity agent, asserting that these interactions are critical in facilitating one’s identity development and academic success.

Indeed, URM faculty provide diverse perspectives in the classroom (e.g., [23,24]). Umbach [9] documented that URM faculty employed active and collaborative learning techniques with greater frequency, emphasizing higher-order cognitive experiences more than white faculty. Higher levels of commitment to teaching were also observed among URM faculty in relation to their white counterparts regardless of their rank [25], which may increase accessibility to faculty for students. Given that “weed out courses” are known to be a major contributing factor to student attrition in STEM, these findings suggest the important roles that faculty, and especially URM faculty, may hold for underrepresented students’ attrition, as well as the attrition of all students in engineering.

It is noteworthy to discuss the effects that URM faculty may have on students’ interracial experiences by creating those contexts that students can engage with one another. Prior research (e.g., [26,27,28] indicated that both white and Black students benefit from interracial dyads when their focus is directed away from prejudice concerns and when interracial interactions are more structured. Similarly, Cole [29] documented that having faculty recommend peer tutoring could offer a practical solution for creating structured interracial contact, as well as opportunities for students to develop intellectually. The presence of URM faculty in this process can be particularly encouraging, as students’ implicit racial biases could be affected by having URM faculty. For example, Lowery et al.’s [30] reported that white participants exhibited lower implicit bias in the presence of a Black experimenter.

At the same time, scholars have suggested that the low representation of URM faculty is one liability in maintaining diversity in student diversity in STEM (e.g., [31]), even though mentoring students should be seen as the responsibility of all faculty members, not just URM faculty. URM faculty can be a role model for underrepresented students as they represent the possibility of one’s group to succeed in that domain [32] and may directly encourage the retention of underrepresented students [33]. Enhanced academic performance and career aspirations were observed when underrepresented students have URM faculty who serve as role models for them [34].

Based on these prior studies, we assume that the presence of URM faculty among engineering faculty can serve as a stepping stone that could facilitate engineering students’ success in engineering. While diversity could be defined in a varied manner, this study is focused on *structural diversity* among faculty. Structural diversity is defined as the numerical representation of diverse racial/ethnic student groups on campus as one of ways of exposing students to diversity in higher education [10]. This type of diversity has been conceptualized as essential, denoted as a particularly strong predictor of engagement with racial diversity for white students [13]. Understanding the effects of faculty demography in smaller units on college campuses (i.e., college and department levels) where students spend more time could provide information regarding the effects of racial and gender diversity in academic disciplines. Some STEM departments notoriously have lower levels of both racial and gender diversity; thus, the subculture of the college can serve as a filter to differentiate students’ levels of immersion into diversity [35]. In sum, this study focused on the proportion of URM faculty within the College of Engineering to understand the effects of structural diversity on engineering students’ perceptions in their career pursuits. Specifically, perceived levels of engineering support and barriers were explored in relation to engineering self-efficacy and the proportion of URM faculty in engineering. Deemer et al. [36] addressed racial composition as a critical factor to impact on STEM students’ stereotype threat, science identity, and STEM self-efficacy. In their work, institutional contexts (i.e., HBCU vs. PWI) were used as a proxy for racial composition. We extended this work by assessing the unique influence that the racial composition of faculty may have on students’ engineering-related social cognitions (self-efficacy), perceived supports, and perceived barriers—all factors that have been linked to academic and career outcomes in STEM fields (i.e., [37]).

### 1.2. Theoretical Framework

In a variety of STEM fields, social cognitive career theory (SCCT; Lent et al., [38,39]) has been served as a prominent theoretical framework to understand persistence among diverse groups of college students as the framework that incorporates contextual variables explicitly (e.g., [40,41,42]).

As the agent role of individuals, self-efficacy, defined as “people’s judgments of their capabilities to organize and execute courses of action required to attain designated types of performances” [43], has been highlighted within the SCCT framework. A consistent and critical role of self-efficacy related to choice goals also has been well established [37,38,44], while displaying correlated relationships with contextual influences, such as perceived supports and barriers [45].

Furthermore, Lent and colleagues [38,39,46] asserted that individuals’ career choice behaviors can be promoted or impeded by one’s personal and environmental context through several different components, such as personal inputs (e.g., race/ethnicity), background contextual affordance (e.g., the access to AP courses), and proximal contextual influences (e.g., perceived barriers and supports in pursuing a specific career). A possible vicarious influence of these individual and environmental factors was construed, while influencing individuals’ socio-cognitive processes directly, indirectly, and interactively. As such, many scholars (e.g., [42,47]) have empirically confirmed the significant influence of perceived barriers and supports on self-efficacy beliefs in engineering. At the same time, most studies have used both perceived barriers and supports as predictors simultaneously on other cognitive variables (e.g., outcome expectations) despite potential complementary or compensatory roles between perceived barriers and supports in the career choice process [37]. A moderate negative relation was found between perceived barriers and supports from the recent meta-analysis by Lent et al. [37], which suggests the needed information by both constructs rather than redundant information. A possible mediating role of perceived supports between perceived barriers to self-efficacy was also discussed that may lessen the level of perceived supports as one experiences of barriers, which, in turn, connect more directly to self-efficacy. Given the importance of a more sophisticated understanding of supports and barriers for addressing underrepresented groups’ career development processes (e.g., [48]), Lent et al. [39] also highlighted a necessity of understanding on the nature of the relationship between perceived supports and barriers.

Indeed, each individual’s phenomenological appraisals may influence how they respond to proximal environmental variables [49]. For example, observing limited in-group role models in the field may reduce one’s level of interests in STEM that can connect to their decision to leave the engineering major. On the other hand, a lack of in-group role models among faculty makes an individual more motivated to become that role model for future generations of students, suggesting the URM faculty’s representation could function as a moderator in the SCCT model.

### 1.3. Present Study

The purpose of the current study was twofold. Our first objective was to extend the SCCT framework by exploring the relationships among perceived engineering barriers, supports, and self-efficacy. As Lent et al. [37] suggested, perceived supports can be enhanced or decreased by the level of perceived barriers and facilitate the development of beliefs in one’s ability [37]. Therefore, it was hypothesized that perceived supports would mediate the relationship between perceived barriers and self-efficacy among a sample of engineering students. Particularly, the mediating relationships between the variables were tested using a longitudinal data set: perceived barriers in engineering at T3 and perceived supports in engineering and engineering self-efficacy at T4. Given that SCCT posited perceived supports and barriers in a more equal model, rather than predicting one another, using data from two points in time, approximately 1 year apart, was likely to add to the current understanding of the three variables’ relationships. A secondary objective of the study was to identify how structural diversity among engineering faculty played a role in the relationships among perceived engineering barriers, supports, and self-efficacy. The proportion of URM faculty within a college of engineering was used as a measure of structural diversity, as a proximal contextual factor, given engineering students’ interactions with the faculty in their college. The SCCT hypothesized that proximal contextual variables interplayed with cognitive and behavioral components both objectively and subjectively, such that greater exposure to URM faculty role models may differentiate the effects of perceived barriers on self-efficacy via perceived supports. Thus, its potential moderating effect on the relationship between perceived engineering supports and engineering self-efficacy was investigated. In summary, as shown in Figure 1, we posited that the relationship between perceived engineering supports and engineering self-efficacy was stronger among students that had a higher proportion of URM faculty to recognize their ability in engineering. Finally, it was hypothesized that there was a larger indirect conditional effect of engineering barriers on self-efficacy through engineering supports when the proportion of URM faculty within the college was low.

## 2. Method

### 2.1. Participants

The initial pool of participants consisted of 841 engineering students attending 11 U.S. universities. However, given the purpose of the current study, participants from six schools that provided their faculty members’ demographic information were included in the study, resulting in a total of 219 engineering student participants for the current study. The sample consisted of 116 (53%) women, 102 (46.6%) men, and 1 (0.5%) transgender individual. The participants identified as Latinx (*n* = 85, 38.8%), White (*n* = 122, 55.7%), multiracial/multiethnic (one group being Latinx or White; *n* = 11, 5%), and other (*n* = 1, 0.5%). Most participants were in their fourth year of college (*n* = 177; 80.8%), with the remaining in their second *(n* = 2; 0.9%), third (*n* = 24; 11.0%), or other years (*n* = 6; 2.7%). The mean age of the sample was 23.3 years (SD = 4.46, range = 21–55). The majors represented were 42 (19.2%) mechanical engineering, 37 (16.9%) computer engineering, 27 (12.3%) electronics engineering, 26 (11.9%) biomedical engineering, 26 (11.9%) chemical engineering, 21 (9.6%) civil engineering, 15 (6.8%) aerospace engineering, 9 (4.1%) industrial engineering, 5 (2.3%) architectural engineering, 2 (0.9%) geosystem engineering and hydrogeology, 2 (0.9%) engineering management, 1 (0.5%) manufacturing engineering, 1 (0.5%) material science and engineering, 1 (0.5%) ocean engineering, 1 (0.5%) cyber operations, 1 (0.5%) material science, 1 (0.5%) robotics engineering, and 1 (0.5%) software engineering.

### 2.2. Procedure

The data were collected through an online survey using Qualtrics during the third and fourth waves of a larger, 5-year longitudinal study of undergraduate engineering majors at 11 U.S. universities in the U.S. Participants were recruited through email announcements, class presentations, and flyers. After consenting to the study, the participants completed a demographic questionnaire and several other engineering-related measures (see Appendix A). The participants received a USD 60 and a USD 80 gift card at T3 and at T4, respectively, for completing the online survey. At the same time, administrators at each participating engineering school/college provided demographic information (race/ethnicity and gender) for the members of their faculty in the same year. Six schools responded with these demographic data, and students from these institutions were included in the current study.

### 2.3. Measures

Engineering Self-Efficacy (ESE; [50]). Engineering self-efficacy was measured with a four-item instrument that assessed the participants’ confidence in one’s ability to succeed in the pursuit of an engineering major (e.g., “excel your engineering major over the next two semesters”) using a 10-point Likert scale ranging from 0 (*completely unsure*) to 9 (*completely sure*). Participants’ responses were averaged with high scores reflecting high levels of engineering self-efficacy. Prior research indicated that engineering self-scores on the measure displayed a positive correlation with engineering outcome expectations, interests, goals, and academic satisfaction in an undergraduate sample of Latinx and White engineering students attending an HSI [51]. Previous studies conducted with engineering self-efficacy also demonstrated good internal consistency of scale scores with coefficient alphas ranging from 0.91 to 0.92 in college student samples [52,53]. The Cronbach’s alpha for the current study was 0.89.

Engineering Perceived Supports and Barriers (ESBS; [50]). Perceived supports and barriers in engineering were measured with a nine-item and a five-item instrument, respectively. Participants responded to the 14 items rated using a five-point scale ranging from 1 (*strongly disagree*) to 5 (*strongly agree*). Support items (e.g., “I get encouragement from friends for pursuing an engineering major”) and barrier items (e.g., “feel pressure from parents or other important people to change your major to some other field”) were summed separately to obtain support and barrier scores, respectively. Item responses were averaged with higher scores indicating higher levels of perceived supports and barriers for pursuing an engineering degree. The adequate internal consistency and validity of scale scores have been demonstrated with engineering student samples [54,55]. The Cronbach’s alpha for the current study was 0.87 and 0.75 for the support and barrier scale scores, respectively.

URM faculty in the College of Engineering. Administrators at each participating engineering school/college were asked to provide demographic information for the faculty in the same academic year in which data were collected from the student participants. Among the 11 universities that participated in the larger project, six responded to this request and provided data on race/ethnicity, gender, and tenure status of the engineering faculty. Given that students’ exposure to URM faculty was the focus of the study, both tenure/tenure track as well as non-tenured faculty were included. The percentage of URM faculty ranged between 31.85% and 66.27% across the six institutions. Five universities were classified as predominantly White institutions, and one was designated a Hispanic-serving institution.

### 2.4. Data Analyses

To test the significance of indirect effect and conditional process effects, two models were tested in the current study: a simple mediation model and a moderated mediation model. Specifically, a simple mediation model was tested to examine whether engineering barriers was associated with engineering self-efficacy via engineering supports, using an application provided by Hayes [56]. An SPSS macro developed by Hayes facilitated the estimation of the indirect effect *ab*, both with a standard theory approach (i.e., the Sobel test) and with a bootstrap approach to obtain confidence intervals (CIs). The stepwise procedure described by Baron and Kenny [57] was also incorporated in the macro.

Next, as depicted in Figure 1, a conditional process model was tested that included the proposed moderator (i.e., the percentage of URM faculty). Hayes [56] described these sorts of conditional process models as a blend of first- and second-stage models, where the associations between a predictor and an intervening variable and between the intervening variable and an outcome are moderated by one or more variables. The macro allowed the recommended bootstrapping methods that probed the significance of the conditional indirect effects at different values of the moderator variable.

## 3. Results

The means, standard deviations, and intercorrelations among the study’s variables for all of the participants are presented in Table 1. An examination of the correlations showed that the perceived engineering barriers at T3 were negatively related to the perceived engineering supports at T4 (*r* = −0.24, *p* < 0.001) but not to engineering self-efficacy at T4 (*r* = −0.12, *p* = 0.08). The results also indicated that the perceived engineering supports at T4 positively correlated with engineering self-efficacy at T4 (*r* = 0.30, *p* < 0.001). However, there was no statistically significant correlation between the percentage of URM faculty and all variables in the model (i.e., perceived engineering barriers and supports and engineering self-efficacy).

Table 2 shows the results of the simple mediation model. The results indicated that the perceived engineering barriers at T3 significantly predicted engineering supports at T4 (*B* = −0.25, *t* = −3.67, *p* = 0.003). The relationship between the perceived engineering supports and engineering self-efficacy at T4 was also significant (*B* = 0.76, *t* = 4.32, *p* < 0.001) while controlling for engineering barriers at T3. However, the perceived engineering barriers at T3 did not have statistically significant associations with engineering self-efficacy at T4, even when controlling the engineering supports at T4 (*B* = −0.13, *t* = −0.75, *p* = 0.46). Finally, an indirect effect of engineering barriers at T3 on engineering self-efficacy at T4 via engineering supports at T4 was significant (Sobel z = −2.754, *p* = 0.005). The results of bootstrapping also confirmed the Sobel test (95% CI [−0.38, −0.06] and a 99% CI [−0.46, −0.03]) for the indirect effect, which did not contain zero.

Next, Table 3 shows the results of the conditional process model. The results indicated that the effect of the interaction between perceived supports in engineering at T4 and the percentage of URM faculty on self-efficacy in engineering at T4 was significant (*B* = −0.05, *t* = −2.16, *p* = 0.032). Thus, conventional procedures were applied for plotting simple slopes (see Figure 2) when the percentage of URM faculty was one standard deviation below and above the mean. The positive indirect associations of perceived supports in engineering and engineering self-efficacy at T4 via perceived barriers in engineering at T3 were significant at low (M − 1 SD) URM faculty percentages (*B* = 1.20, *B_SE_* = 0.23, 95% CI [0.74, 1.67]), as well as at mean (M) URM faculty percentages (*B* = 0.81, *B_SE_* = 0.16, 95% CI [0.49, 1.12], while no significant association was observed at high (M + 1 SD) URM faculty percentages.

Although the results indicated that the interaction effect between the percentage of URM faculty and perceived engineering supports at T4 influenced engineering self-efficacy at T4, they did not directly assess the conditional indirect effects model depicted in Figure 1. Therefore, a more detailed examination of the conditional indirect effects model was examined at three percentage points of the URM faculty (see middle of Table 3): the mean (40.40%), one standard deviation above the mean (48.95%), and one standard deviation below the mean (31.85%). Normal-theory tests indicated all three conditional indirect effects were significantly positive, and bootstrap supported these results not including zero in the values. In other words, the indirect and stronger effect of the perceived engineering barriers on engineering self-efficacy through perceived engineering supports was observed when the percentage of URM faculty was low. Furthermore, as shown in the bottom section of Table 3, when the range of URM faculty percentages was expanded, the positive effect turned out to be non-significant (i.e., when the percentage of URM faculty was 48.94%, the indirect effect of perceived barriers on self-efficacy was not significant with *p* = 0.10). Thus, the data suggested that high levels of perceived engineering barriers at T3 had no effect on increasing perceived engineering supports at T4 when URM faculty comprised approximately 50% of the entire engineering faculty at an institution.

## 4. Discussion

The current study extended the SCCT literature in the domain of engineering by examining the impacts of engineering faculty racial/ethnic composition on social cognitive outcomes among a sample of diverse racial/ethnic engineering college students. By integrating the proportion of URM faculty as a moderating variable in the SCCT model, our findings indicated that both perceived engineering barriers and perceived engineering supports significantly related to the perceived self-efficacy even after controlling for the effects of the other. Specifically, perceived engineering barriers were negatively associated with self-efficacy when controlling for engineering support. Perceived engineering supports were positively associated with self-efficacy when controlling engineering barriers, and perceived engineering supports mediated the effects of engineering barriers on self-efficacy. Finally, the proportion of URM faculty moderated the mediation effects of engineering supports between engineering barriers and self-efficacy; when the proportion of URM faculty reached a certain level, high levels of perceived engineering barriers had no effect on increasing perceived engineering supports. Below, we expand on these findings from the study.

Congruent with SCCT propositions and similar to those of prior studies among college students (e.g., [39,58,59,60]), our findings supported the relations between environmental factors and engineering self-efficacy. Lent and colleagues [39] posited that social environmental factors impacted one’s self-efficacy and academic outcomes and found empirical support for the proposed relationship between social environmental factors and self-efficacy in a meta-analysis of STEM studies [37]. Although prior studies with college student samples and engineering college students have already found significant relations between contextual factors and self-efficacy, the findings from this study deepened our understanding of this relationship in important ways. First, by controlling for the effect of each contextual factor, we found that perceived supports and barriers in engineering both had unique effects on engineering self-efficacy after controlling the other. In other words, perceived barriers and supports in engineering should not be taken as two sides of a coin. Instead, they were two closely related but distinct constructs, and each had a unique impact on social cognitions (i.e., self-efficacy). Furthermore, this study added to our knowledge of the relationship among contextual supports, contextual barriers, and self-efficacy by revealing the mediation effects of perceived engineering support between perceived engineering barriers and engineering self-efficacy. This is a unique contribution of the current study, given that previous studies mostly focused on the direct impacts of contextual barriers and contextual support on self-efficacy but failed to attend to the mediation role of contextual supports (e.g., [60]).

Another novel component of the current study was the inclusion of structural diversity among the engineering faculty as a contextual variable. This variable captured the systematic phenomenon that students encountered on a daily basis. Although scholars have explored faculty roles in higher education, most studies have focused on student–faculty racial/ethnic match or having a faculty member of the same race/ethnicity (e.g., [61,62]). In the present study, we attempted a systematic exploration of the numeric representation of URM faculty within engineering colleges. This allowed us to discuss systematic and practical environmental interventions and solutions. While there have been studies that observed the influence of aggregated demographic composition on students’ perceptions of stereotype threat in STEM across institutional types (i.e., HBCU vs. PWI) (e.g., [36]), to our knowledge, this study is the first to collect institutional-level URM faculty composition data to examine their impact on individual-level outcomes among engineering students. Considering that the presence of URM faculty can affect underrepresented students’ persistence in degree pursuits, the present study provides some answers for scholars that question the formal and informal systems within engineering education itself that may create obstacles for underrepresented students (e.g., [63,64]).

Importantly, the conditional indirect effects between perceived barriers and self-efficacy through perceived supports were observed to decrease as the proportion of URM faculty increased. In other words, engineering students’ perceived barriers had no meaningful effect on decreasing perceived supports in engineering when the proportion of URM faculty reached a certain level (48%). The results of this research indicated that the proportion of URM faculty was a contextual factor that influenced students’ perceptions of barriers and supports, which in turn were influential in developing beliefs in their ability to pursue engineering degrees. In university settings, this result suggests that the proportion of URM faculty could be a significant factor in decreasing the harmful effects of perceived engineering barriers among students.

By obtaining both Latinx and white engineering student participants, this study highlighted the benefits that faculty diversity could bring in higher education for all students. Much of the research suggested that the effects of student–faculty interaction were “conditional” [65,66,67]—the beneficial influence of student–faculty interaction on student outcomes differed based on students’ characteristics such as gender and race/ethnicity. Given the well-documented privilege of White students, most scholars have examined the effects of URM faculty for underrepresented students as in-group role models (e.g., [68,69]). However, the current study showed universal benefits that a high proportion of racially diverse engineering faculty could have on engineering students’ self-efficacy regardless of students’ racial/ethnic group status. Even though further research is required, including the internal mechanism through which different groups of students may interact with diverse faculty, this finding demonstrated an interplay with perceived barriers, perceived supports, and engineering self-efficacy, as a proximal contextual factor.

### 4.1. Implications for Practice and Future Research

Given that STEM knowledge has been increasingly a key input to production in the marketplace; accounting for nearly one-third (31%) of global output, as well as leading the major economies at 38% of their gross domestic products, scholars have been motivated to identify effective interventions that prepare students to enter these challenging and important fields of study (e.g., [70]). As a result; considerable research has been conducted that emphasizes the strategies that students might adopt to successfully navigate engineering, focusing on students’ personal levels (e.g., backgrounds or characteristics) by developing program-level interventions (e.g., [71]). However; scholars have pointed out the need for systematic approaches to understand students’ barriers and systems-tailored interventions [72], focusing on structural levels, such as institutions, environments, and culture that may exclude groups of people. The present study contributed to systematic approaches that explore students’ perceived barriers by analyzing the environments and students’ experiences within these environments, aiming to focus the discussion on structural changes that ultimately support engineering students’ engagement in their academic and career development. Given that a number of person-level factors, such as a lack of self-confidence in STEM subjects, low STEM career self-efficacy, and lack of social support and encouragement to pursue STEM-related educational and career goals, have been considered as exemplary attributions for low participation in STEM [73,74], the findings of the current study suggest different avenues for interventions targeted at increasing the self-efficacy of engineering college students: recruiting and retaining significant concentrations of URM faculty in engineering. While the presence of URM faculty mostly has been considered to be a benefit for underrepresented students, this study demonstrated that engineering URM faculty could benefit all students regardless of their racial/ethnic background. Thus, it would behoove engineering college deans, department chairs, and engineering faculty to reconsider academic environments within engineering schools/colleges.

Although the number of diverse faculty at U.S. colleges and universities is slowly growing [75], it is questionable whether university structures, policies, and procedures have grown in parallel to support them. In fact, IDE policies and initiatives are under attack and being challenged in several states today. This legislation poses challenges to recruiting and retaining racially diverse faculty, and based on our findings, the career development of students at institutions in states that have banned or restricted IDE efforts may be impacted by the lack of diversity among engineering faculty. URM faculty remain vulnerable to continued tokenization, microaggression, and subtle discrimination [76,77]. Salary disparities in white faculty, as well as obstacles to leadership for URM faculty, have been identified [78]. URM faculty tend to be isolated in their work environments and carry disproportionate advising and service loads for underrepresented students, resulting in a fragile pipeline for URM faculty (e.g., [79,80]). To increase URM faculty, Hughes [81] suggested looking at intentional actions, strategic planning, and relationships. Specifically, at the institutional level, institutions should consider partnering with URM faculty within their university communities to construct plans and steps to create equitable and nondiscriminatory spaces where URM faculty feel welcomed and valued. Shavers et al. [82] also highlighted the importance of agency of URM faculty for implementing these initiatives at the times and methods of their choosing. These processes are encouraged within the college and department levels as well. For example, administrators and academic professors in engineering could engage in continuous discussions, such as reviewing policies and updating them to reflect anti-discriminatory practices. Providing appropriate training for faculty serving on search committees is also suggested. Cavanaugh and Green [83] argued the needs for having internal faculty consultants to serve on search committees to focus on eliminating bias and inequities that affect the search process to improve racial and ethnic diversity.

College personnel who develop and evaluate STEM academic and career interventions, particularly for underrepresented groups in engineering such as Latinx and women, can consider ways to apply these findings to help high school and college students develop engineering self-efficacy. Providing academic support for engineering college students from underrepresented groups by connecting students with engineering URM faculty at their institution could potentially help to increase their self-efficacy specific to their engineering study. College personnel can also work with high school and college students to improve their self-efficacy in the STEM field by analyzing potential environmental factors that are related to their engineering self-efficacy. Interventions focusing on identifying support from various resources and figuring ways to lower experienced and expected barriers might also empower these youth to pursue STEM majors. Psychoeducation could be incorporated in STEM courses and college advising to inform students that self-efficacy could be related to environmental factors such as faculty composition. This could especially be empowering for underrepresented groups of students in STEM fields to help them gain deeper insight into their self-perceptions and academic experiences as members of underrepresented groups in STEM.

### 4.2. Limitations and Conclusions

The findings of this study should be considered in the context of the study’s limitations. First, limited representations of the sample may limit our understanding of relations among the variables in the model. Specifically, our sample only included Latinx and white engineering students. The current study’s sample was drawn from the larger project that focused on Latinx and white-identified engineering students. Although we were able to shed light on how faculty diversity could benefit both groups of students, future studies should consider including other underrepresented groups of students that could provide further insight to understand the model in this study. Similarly, the majority of participants in the current study were in the latter years of college and represent a group that has persisted in the field, at least through their college education. This could limit the generalizability of our findings given that the year in college could impact college students’ perceptions about themselves and their social environmental factors (Flores et al., 2014 [51]). Future research should control years of study as a factor that may impact college experiences and recruit a more evenly distributed college student sample to examine the model in this study. Another limitation was that we only included engineering self-efficacy as the outcome variable. According to SCCT, proximal contextual factors are related to many other social cognitive factors, such as interests, positive and negative outcome expectations, and academic persistence. Relatedly, other factors should be considered to predict engineering self-efficacy. For example, institutional types that connect to peer diversity have been demonstrated as an influential component on students’ college experiences and major pursuits (e.g., [84]). The broader impacts of faculty composition, as well as factors to shape self-efficacy, could be targeted to provide more avenues for improving college experiences for engineering students from unrepresented groups. Last, our study included Asian/Asian American-identified faculty as URM faculty. It was an intentional effort to dilute the harmful stereotype toward Asian/Asian American groups in STEM fields as they also experience systemic racial disparities, such as consistent underfunding by the flagship U.S. agency [85] and being the least likely group in the U.S. to be promoted to management [86]. Given the overrepresentation of Asian/Asian American groups within engineering [87], however, future research may consider exploring influences on students’ social cognitions by excluding Asian/Asian American faculty in engineering. This approach, though, should be implemented with caution as not all Asian groups are overrepresented in engineering (e.g., [88,89]).

The purpose of the current study was to examine the impacts of contextual factors (i.e., perceived engineering barriers, perceived engineering supports, and the proportion of URM faculty) on engineering self-efficacy among a diverse group of engineering college students. One major finding of this study was the unique role of both the perceived engineering barriers and supports on engineering self-efficacy and the mediation role of engineering supports on the relationship between engineering barriers and engineering self-efficacy. The contribution of the current study is unique as it incorporates a broader systematic perspective, by including the proportion of URM faculty and demonstrating that having a diverse faculty can benefit all students. Another major finding was the moderating role of faculty composition in the relationship between contextual factors and engineering self-efficacy. Future research can build on these findings by examining the longitudinal impacts of social environmental factors on engineering self-efficacy among a diverse college student sample. Further, future research that examines the influence of a diverse faculty can incorporate more social cognitive factors (e.g., interest and academic satisfaction) to assess the broader impacts of faculty composition on the academic and career development of college students.

## Figures and Tables

**Figure 1 behavsci-14-00564-f001:**
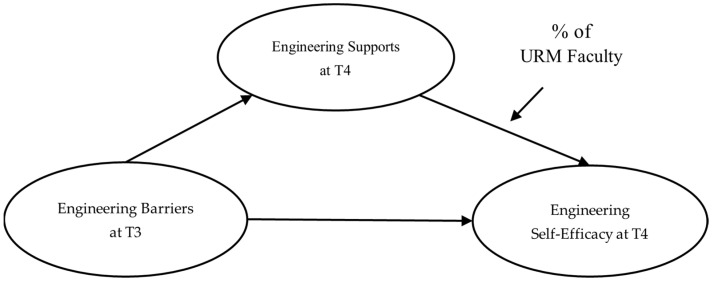
Theoretical model.

**Figure 2 behavsci-14-00564-f002:**
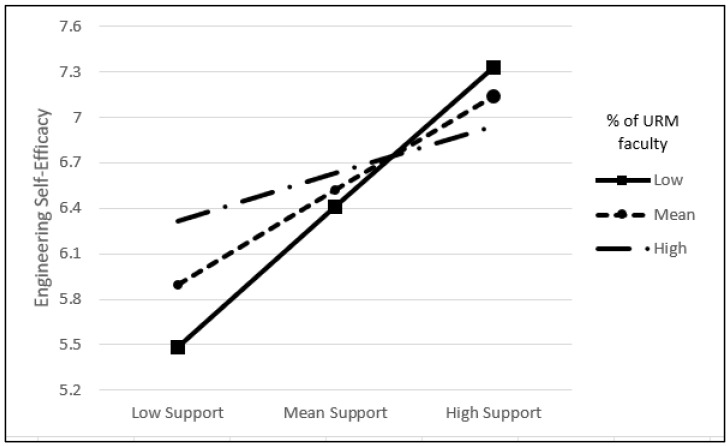
Interaction effect of % of REM faculty between engineering barriers and engineering supports.

**Table 1 behavsci-14-00564-t001:** Mean values and correlations among the study’s variables.

Variables	*M* (SD)	1	2	3	4
1. Engineering barriers at T3	1.99 (0.77)	-			
2. Engineering supports at T4	3.71 (0.78)	−0.24 ***	-		
3. Engineering self-efficacy at T4	6.56 (2.06)	−0.12	0.30 ***	-	
4. % of URM Eng faculty members	40.40 (8.62)	0.09	0.01	0.02	-

Note. *n* = 254. Standard deviations are in parentheses. URM = underrepresented racial minority; Eng = engineering. *** *p* < 0.001.

**Table 2 behavsci-14-00564-t002:** Regression results for simple mediation.

Variable	*B*	*SE*	*t*	*z*	*p*	95% CI
Direct and total effects						
Engineering self-efficacy at T4 regressed on engineering barriers at T3	−0.320	0.180	−1.772		0.078	
Engineering supports at T4 regressed on engineering barriers at T3	−0.245	0.067	−3.666		0.003	
Engineering self-efficacy at T4 regressed on engineering Supports at T4, controlling for engineering barriers at T3	0.760	0.176	4.317		<0.001	
Engineering self-efficacy at T4 regressed on engineering barriers at T3, controlling for engineering supports at T4	−0.133	0.179	−0.746		0.457	
Indirect effect (IE)		0.068		−2.752		[−0.319, −0.054]
Bootstrap results for IE		0.084		−0.186		[−0.381, −0.058]

Note. *n* = 219; unstandardized regression coefficients are reported. Bootstrap sample size = 5000. CI = confidence interval. With significance using normal distribution. IE = −0.186, 99% CI = [−0.464, −0.033].

**Table 3 behavsci-14-00564-t003:** Regression results for conditional indirect effects.

Variables	*B*	*SE*	*t*	*p*	IE	*SE*	*z*	*p*	95% Bias-Corrected Bootstrap 95% CI
	Engineering Supports at T4								
Constant	0.489	0.143	3.420	0.001					
Engineering barriers at T3	−0.245	0.067	−3.666	0.000					
	Engineering Self-Efficacy								
Constant	6.819	0.382	17.864	0.000					
Engineering barriers at T3	−0.128	0.179	−0.711	0.478					
Engineering supports at T4	0.756	0.176	4.301	0.000					
% of URM faculty	0.006	0.016	0.409	0.683					
ES X % of URM faculty	−0.046	0.021	−2.157	0.032					
	Conditional Indirect Effect at Specified Values % of URM faculty								
% of URM faculty									
−1 SD (31.85%)					1.076	0.259	4.158	0.000	[0.566, 1.586]
M (40.40%)					0.756	0.176	4.301	0.000	[0.410, 1.103]
+1 SD (48.95%)					0.434	0.262	1.657	0.099	[−0.082, 0.950]
	Conditional Indirect Effect at Range of % of URM faculty								
% of URM faculty									
33.57					1.012	0.232	4.360	0.000	
35.29					0.948	0.209	4.533	0.000	
37.01					0.883	0.191	4.624	0.000	
38.73					0.819	0.180	4.561	0.000	
40.46					0.754	0.176	4.290	0.000	
42.18					0.690	0.181	3.822	0.000	
43.90					0.626	0.193	3.243	0.001	
45.62					0.561	0.212	2.652	0.009	
47.34					0.497	0.235	2.112	0.036	
47.83					0.478	0.243	1.971	0.050	

Note. Unstandardized regression coefficients are reported. Bootstrap sample size = 5000; CI = confidence interval; IE = indirect effect.

## Data Availability

The data that support the findings of this study are available on request from the corresponding author, Bo Hyun Lee.

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
