# Peer review of "The Role of Contextual Variables and Structural Diversity on College Students’ Engineering Self-Efficacy"

_behavsci, 2024, doi:10.3390/bs14070564_

Round 1

Reviewer 1 Report

Comments and Suggestions for Authors

  The study implemented the concept of structural diversity on faculty in higher education. The proportion of faculty of color within the College of Engineering as a moderating variable is examined within social cognitive career theory (SCCT) model. The participants were 254 diverse engineering students from six universities. |The findings were that both perceived engineering barriers and perceived engineering supports are relate to perceived self-efficacy, and perceived engineering supports mediated the effects of engineering barriers on self-efficacy. Besides, a moderated mediation effect of the proportion of faculty of color was tested, comparing universities with hight and low proportions.

 This is simple routine research tested a hypothesis related to the effect of the proportion of faculty of color. The relationships among the perceived engineering barriers, perceived engineering supports and perceived self-efficacy, presented via a path analysis , while the novel part is the addition of a variable expressing the proportion of faculty of color, which is found to impose a moderated mediation effect.

Even though the research is short, the paper is well written, providing well informed relative literature and supporting the research questions. The statistical part is also well performed. The discussion section is adequate developed including the limitations and suggestion for future research

In my opinion, the paper is worth publishing, in this form.

Comments on the Quality of English Language

  The study implemented the concept of structural diversity on faculty in higher education. The proportion of faculty of color within the College of Engineering as a moderating variable is examined within social cognitive career theory (SCCT) model. The participants were 254 diverse engineering students from six universities. |The findings were that both perceived engineering barriers and perceived engineering supports are relate to perceived self-efficacy, and perceived engineering supports mediated the effects of engineering barriers on self-efficacy. Besides, a moderated mediation effect of the proportion of faculty of color was tested, comparing universities with hight and low proportions.

 This is simple routine research tested a hypothesis related to the effect of the proportion of faculty of color. The relationships among the perceived engineering barriers, perceived engineering supports and perceived self-efficacy, presented via a path analysis , while the novel part is the addition of a variable expressing the proportion of faculty of color, which is found to impose a moderated mediation effect.

Even though the research is short, the paper is well written, providing well informed relative literature and supporting the research questions. The statistical part is also well performed. The discussion section is adequate developed including the limitations and suggestion for future research

In my opinion, the paper is worth publishing, in this form.

Author Response

Comments 1: The study implemented the concept of structural diversity on faculty in higher education. The proportion of faculty of color within the College of Engineering as a moderating variable is examined within social cognitive career theory (SCCT) model. The participants were 254 diverse engineering students from six universities. |The findings were that both perceived engineering barriers and perceived engineering supports are relate to perceived self-efficacy, and perceived engineering supports mediated the effects of engineering barriers on self-efficacy. Besides, a moderated mediation effect of the proportion of faculty of color was tested, comparing universities with high and low proportions.

Response 1: Thank you for your summary on our study!

Comments 2: This is simple routine research tested a hypothesis related to the effect of the proportion of faculty of colorThe relationships among the perceived engineering barriers, perceived engineering supports and perceived self-efficacy, presented via a path analysis, while the novel part is the addition of a variable expressing the proportion of faculty of color, which is found to impose a moderated mediation effect.

Response 2: We appreciate your recognition of our novel findings!

Comments 3: Even though the research is short, the paper is well written, providing well informed relative literature and supporting the research questions. The statistical part is also well performed. The discussion section is adequate developed including the limitations and suggestion for future research. In my opinion, the paper is worth publishing, in this form.

Response 3: We truly thank your sharing. Hope our revision has strengthened our findings. 

Reviewer 2 Report

Comments and Suggestions for Authors

Thanks for this important work! I especially found it interesting and useful to show that overall demographics of racial diversity in the faculty helped all students regardless of race.

A few suggestions for improvement:

1. In lit review section 1.2, a figure schematic to illustrate SCCT would be useful, especially to set up the relationships you are statistically testing. Connecting SCCT to your theoretical model in Figure 1 is necessary - I don't quite see the connection. Are you only connecting to self-efficacy in SCCT or any of the other constructs?

2. Providing your entire survey with all items for each scale would facilitate better understanding of the study - this could be included in supplemental information.

3. On line 267 you say REM faculty for the first time. It would be good to spell out what this stands for in the text there in addition to where you have it in figure legend in line 290.

Author Response

Comments 1: In lit review section 1.2, a figure schematic to illustrate SCCT would be useful, especially to set up the relationships you are statistically testing. Connecting SCCT to your theoretical model in Figure 1 is necessary - I don't quite see the connection. Are you only connecting to self-efficacy in SCCT or any of the other constructs?

Response 1: We appreciate your suggestion. We added additional description how we developed the tested model as follows: 

In-text: Furthermore, Lent and colleagues (1994, 2000, 2008) asserted that individuals’ career choice behaviors can be promoted or impeded by one’s personal and environ-mental context through several different components, such as personal inputs (e.g., race/ethnicity), background contextual affordance (e.g., the access to AP courses), and proximal contextual influences (e.g., perceived barriers and supports in pursuing a specific career). A possible vicarious influence of these individual and environmental factors was construed, while influencing individuals’ socio-cognitive processes directly, indirectly, and interactively. As such, many scholars (e.g., Lent et al., 2011; Raelin et al., 2014) have empirically confirmed the significant influence of perceived barriers and supports on self-efficacy beliefs in engineering. At the same time, most studies have used both perceived barriers and supports as predictors simultaneously on other cog-nitive variables (e.g., outcome expectations) despite of potential complementary or compensatory roles between perceived barriers and supports in career choice process (Lent et al., 2018). A moderate negative relation was found between perceived barriers and supports from the recent meta-analysis by Lent et al. (2018), which suggests the needed information by both constructs rather than redundant information. A possible mediating role of perceived supports between perceived barriers to self-efficacy was also discussed that may lessen the level of perceived supports as one experiences of barriers which, in turn, connect more directly to self-efficacy. Given the importance of a more sophisticated understanding of supports and barriers for addressing underrepresented groups’ career development process (e.g., McWhirter, 1997), Lent et al. (2000) also highlighted a necessity of understanding on nature of the relationship be-tween perceived supports and barriers.

As Lent et al. (2018) suggested that perceived supports can be enhanced or decreased by the level of perceived barriers and facilitate the development of beliefs in one’s abil-ity (Lent et al., 2018). Therefore, it is hypothesized that perceived supports will mediate the relationship between perceived barriers and self-efficacy among a sample of engineering students. Particularly, the mediating relationships between the variables will be tested using longitudinal data set: Perceived barriers in engineering at T3, and perceived supports in engineering and engineering self-efficacy at T4. Given that SCCT posits perceived supports and barriers in a more equal model, rather than predicting one another, using data from two points in time, approximately 1 year apart, is likely to add to the current understanding of the three variables’ relationships. A secondary objective of the study is to identify how structural diversity among engineering faculty plays a role in the relationships among perceived engineering barriers, supports, and self-efficacy. The proportion of URM faculty within a college of engineering is used as a measure of structural diversity, as a proximal contextual factor, given engineering students’ interactions with faculty in their college. SCCT hypothesizes that proximal contextual variables interplay with cognitive and behavioral components both objec-tively and subjectively, such that greater exposure to URM faculty role models may differentiate the effect of perceived barriers on self-efficacy via perceived supports. Thus, its potential moderating effect on the relationship between perceived engineering supports and engineering self-efficacy is investigated. In summary, as shown in Figure 1, we posit that the relationship between perceived engineering supports and engineering self-efficacy is stronger among students that have a higher proportion of URM faculty to recognize their ability in engineering. Finally, it is hypothesized that a larger indirect conditional effect of engineering barriers on self-efficacy through engineering supports when the proportion of URM faculty within the college is low.

Comments 2: Providing your entire survey with all items for each scale would facilitate better understanding of the study - this could be included in supplemental information.

Response 2: We appreciate your suggestion. As you requested, we included the entire survey items in the supplemental information.

Comments 3: On line 267 you say REM faculty for the first time. It would be good to spell out what this stands for in the text there in addition to where you have it in figure legend in line 290.

Response 3: We thank your attention. Based on editor’s comment, we switched terminology into URM faculty consistently throughout the manuscript. We made sure it is spelled out when using it for the first time.

In-text: In STEM, Price (2010) documented a positive correlation by Black identified students between the persistence and the number of STEM courses taught by a Black instructor, suggesting an integral role of underrepresented racial minority (URM) faculty in helping to shape underrepresented students’ sense of belonging through their interactions (Tinto, 2017).
